# Phenome-wide analyses establish a specific association between aortic valve *PALMD* expression and calcific aortic valve stenosis

Zhonglin Li[1], Nathalie Gaudreault [1], Benoit J. Arsenault[1,2], Patrick Mathieu [1,3], Yohan Bossé [1,4] & Sébastien Thériault [1,5✉]

Calcific aortic valve stenosis (CAVS) is a frequent heart disease with significant morbidity and mortality. Recent genomic studies have identified a locus near the gene *PALMD* (palmdelphin) strongly associated with CAVS. Here, we show that genetically-determined expression of *PALMD* in the aortic valve is inversely associated with CAVS, with a stronger effect in women, in a meta-analysis of two large cohorts totaling 2359 cases and 350,060 controls. We further demonstrate the specificity of this relationship by showing the absence of other significant association between the genetically-determined expression of *PALMD* in 9 tissues and 852 phenotypes. Using genome-wide association studies meta-analyses of cardiovascular traits, we identify a significant colocalized positive association between genetically-determined expression of *PALMD* in four non-cardiac tissues (brain anterior cingulate cortex, esophagus muscularis, tibial nerve and subcutaneous adipose tissue) and atrial fibrillation. The present work further establishes *PALMD* as a promising molecular target for CAVS.

[1] Institut Universitaire de Cardiologie et de Pneumologie de Québec-Université Laval, Quebec City, QC G1V 0A6, Canada. [2] Department of Medicine, Laval University, Quebec City, QC G1V 0A6, Canada. [3] Department of Surgery, Laval University, Quebec City, QC G1V 0A6, Canada. [4] Department of Molecular Medicine, Laval University, Quebec City, QC G1V 0A6, Canada. [5] Department of Molecular Biology, Medical Biochemistry and Pathology, Laval University, Quebec City, QC G1V 0A6, Canada. ✉email: sebastien.theriault@criucpq.ulaval.ca

Calcific aortic valve disease (CAVS) is the most common valvular heart disease, with a prevalence of about 2% after 65 years old[1,2]. CAVS is a progressive disease leading to decreased exercise capacity, heart failure and death[3]. There is currently no medical therapy to prevent the progression of CAVS[4]. The only treatments available for symptomatic patients are either surgical or by transcatheter valve replacement. These procedures are associated with a significant morbidity/mortality rate and elevated cost[5]. Thus, finding new molecular targets is an urgent priority in order to develop pharmacological treatments, which could prevent or slow the progression of CAVS[6]. The robust association between variants at the *LPA* locus and CAVS risk is promising for the use of lipoprotein (a)-lowering agents to treat CAVS, which are currently under evaluation[7]. More recently, a second locus associated with CAVS, *PALMD*, was identified[8,9]. Using an expression quantitative trait loci (eQTL) dataset in human aortic valves, we showed that a decrease of *PALMD* expression in the valve is associated with CAVS risk and that this relationship is likely causal[9]. Of note, the variants associated with CAVS did not affect *PALMD* expression in 44 other tissues in the Genotype-Tissue Expression (GTEx) project. *PALMD* is expressed with variable abundance in most human tissues. The biological function of PALMD is largely unknown, but investigations suggest that it is localized in the cytosol where it is expected to play a role in plasma membrane dynamics and myogenic differentiation[10,11]. Agents modulating *PALMD* expression or a related pathway in the aortic valve could represent a promising avenue for the treatment of CAVS. However, the potential implication of *PALMD* in other systems and health conditions remains unknown.

The availability of comprehensive phenotypic data from electronic health records, such as in the UK Biobank[12], has made possible the use of phenome-wide association studies (pheWAS) to examine the impact of one or many genetic variants across a broad range of human phenotypes[13,14]. These can include vital signs, physical traits, health conditions (using a standardized disease classification), surgical procedures, imaging and laboratory test results. A pheWAS is an effective tool for the prioritization of drug targets and drug repositioning since it allows the prediction of the beneficial, as well as adverse effects of the modification of a specific molecular target[15]. The approach can be considered as a long-term randomized trial performed by nature, where the genetic variants mimic a potential intervention. Recent advances in the field of transcriptomic analyses have made possible the prediction of gene expression in different tissues from genotypes[16-18]. This allows to go one step further in target prioritization and evaluate the association of the level of gene expression in a specific tissue with a wide range of health conditions.

To determine the therapeutic potential of modulating *PALMD* expression, we investigated the effect of *PALMD* genetically-determined expression in several tissues, including the aortic valve, on multiple health conditions using a phenome-wide approach. We show that the association with CAVS is specific to the aortic valve and that *PALMD* expression in other tissues could be associated with atrial fibrillation.

## Results

**Gene expression in the aortic valve and risk of CAVS.** *PALMD* genetically-determined expression in the aortic valve was estimated in the UK Biobank and QUEBEC-CAVS. The model included 17 SNPs located within 1 Mb of *PALMD* and explained 48% of the expression variance (Table 1 and Supplementary Data 1). Predicted expression in the aortic valve was inversely associated with CAVS in the UK Biobank (OR = 0.84 [0.80–0.89]

**Table 1 Number of variants included and proportion of variance explained in the models used to predict *PALMD* expression in 9 tissues.**

| Tissue | Number of variants | $R^2$ | Number of samples |
|---|---|---|---|
| Aortic valve | 17 | 0.480 | 233 |
| Brain anterior cingulate cortex | 15 | 0.059 | 102 |
| Cells transformed fibroblasts | 8 | 0.023 | 256 |
| Esophagus gastroesophageal junction | 16 | 0.050 | 185 |
| Esophagus mucosa | 2 | 0.040 | 307 |
| Esophagus muscularis | 9 | 0.033 | 287 |
| Nerve tibial | 49 | 0.040 | 305 |
| Pancreas | 42 | 0.199 | 180 |
| Subcutaneous adipose tissue | 24 | 0.050 | 328 |

$R^2$: Proportion of expression variance explained in the model.

per SD, $P = 9.3 \times 10^{-10}$), in QUEBEC-CAVS (OR = 0.83 [0.76–0.91] per SD, $P = 7.4 \times 10^{-5}$) and in a meta-analysis combining the two cohorts (OR = 0.84 [0.80–0.88] per SD, $P = 1.1 \times 10^{-12}$) (Fig. 1 and Supplementary Table 1). There was a strong probability of colocalization (PP4 = 99.7%) (Fig. 2).

In sex-stratified analyses, the association between genetically-determined *PALMD* expression in the aortic valve and CAVS was stronger in women compared to men (women OR = 0.76 [0.70–0.83] per SD; men OR = 0.88 [0.83–0.94] per SD, $P_{het}$ = 0.0048; Fig. 1 and Supplementary Table 1) in a meta-analysis combining the UK Biobank and QUEBEC-CAVS. Of note, the gene expression dataset in aortic valves used for prediction included individuals of both sexes (117 women and 116 men). *PALMD* expression was similar in both sexes ($P = 0.20$) and there was no difference in the effect of the strongest SNP in the model, rs6702619, on expression according to sex ($P_{het}$ = 0.36; Fig. 3).

**PheWAS of genetically-determined expression in aortic valve**. We evaluated the association between *PALMD* genetically-determined expression in the aortic valve and 852 phenotypes (Supplementary Data 2) in 353,378 European individuals from the UK Biobank. There were significant associations with aortic valve disorders (ICD10 I35 code, OR = 0.86 [0.82–0.90] per SD, $P = 2.6 \times 10^{-12}$) and aortic valve replacement or repair (OPCS4 codes K26 or K302, OR = 0.81 [0.76–0.86] per SD, $P = 4.9 \times 10^{-11}$) (Fig. 4 and Supplementary Data 3). Other than diagnosis and procedures for aortic valve stenosis, there was no phenotype significantly associated with genetically-determined *PALMD* expression in the aortic valve in the UK Biobank after correction for multiple testing. A pheWAS analysis restricted to the lead CAVS risk variant, rs6702619, yielded similar results (Supplementary Fig. 1 and Supplementary Data 4).

**Transcriptome-wide eQTL analysis of lead CAVS risk variant.** Transcriptome-wide eQTL analyses for rs6702619 in 48 tissues included in the GTEx project v7 European samples, as well as in our aortic valve expression dataset confirmed the known signal for *PALMD* expression in the aortic valve. No other association was significant following correction for multiple testing (Supplementary Data 5).

**PheWAS of genetically-determined expression in other tissues**. We then used models derived from the GTEx project to estimate genetically-determined expression of *PALMD* in other tissues. Weights were available for 8 tissues: brain anterior cingulate

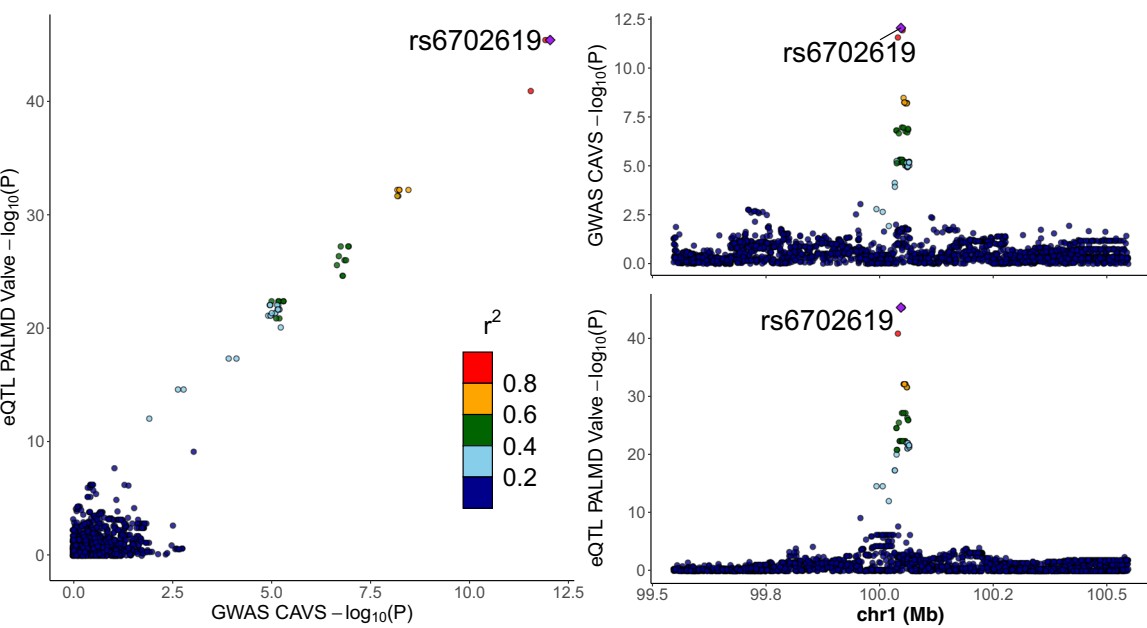

| Women | OR [95% CI] | P value | | N total | N cases | N ctl |
|---|---|---|---|---|---|---|
| UK Biobank | 0.77 [0.70, 0.85] | 1.1e−07 | | 187369 | 451 | 186918 |
| QUEBEC−CAVS | 0.75 [0.64, 0.87] | 0.00022 | | 724 | 367 | 357 |
| **Meta analysis** | **0.76 [0.70, 0.83]** | **2.2e 10** | | **188093** | **818** | **187275** |

| Men | OR [95% CI] | P value | | N total | N cases | N ctl |
|---|---|---|---|---|---|---|
| UK Biobank | 0.88 [0.82, 0.94] | 0.00019 | | 163024 | 899 | 162125 |
| QUEBEC−CAVS | 0.89 [0.80, 0.99] | 0.037 | | 1302 | 642 | 660 |
| **Meta analysis** | **0.88 [0.83, 0.94]** | **4.0e 05** | | **164326** | **1541** | **162785** |

| All | OR [95% CI] | P value | | N total | N cases | N ctl |
|---|---|---|---|---|---|---|
| UK Biobank | 0.84 [0.80, 0.89] | 9.3e−10 | | 350393 | 1350 | 349043 |
| QUEBEC−CAVS | 0.83 [0.76, 0.91] | 7.4e−05 | | 2026 | 1009 | 1017 |
| **Meta analysis** | **0.84 [0.80, 0.88]** | **1.1e 12** | | **352419** | **2359** | **350060** |

**Fig. 1 *PALMD* genetically-determined expression in the aortic valve and CAVS risk in women and men.** Forest plot showing the association between *PALMD* genetically-determined expression in the aortic valve and CAVS according to sex in UK Biobank and QUEBEC-CAVS. OR: Odds ratios for CAVS per SD increase in genetically-determined expression of *PALMD* in the aortic valve.

**Fig. 2 Aortic valve *PALMD* eQTL and GWAS association with CAVS.** LocusCompare plot[30] showing the relationship between aortic valve *PALMD* eQTL and GWAS association with CAVS for variants located within 1 Mb of *PALMD*. Colocalization PP4 = 99.7%. GWAS association was obtained from a meta-analysis of QUEBEC-CAVS and UK Biobank. The lead GWAS variant is annotated.

cortex, transformed fibroblasts, gastroesophageal junction, esophagus mucosa, esophagus muscularis, tibial nerve, pancreas, subcutaneous adipose tissue (Supplementary Data 1). The models explained between 2 and 20% of the gene expression variance (Table 1). In pheWAS of *PALMD* genetically-determined expression in these tissues including 852 phenotypes, we did not observe any significant association after correction for multiple

testing (Supplementary Fig. 2 and Supplementary Data 6–13). The probability of colocalization between *PALMD* genetically-determined expression and CAVS was low for these 8 tissues (all PP4 < 5%) (Supplementary Fig. 3).

**Impact of genetically-determined expression on cardiovascular phenotypes.** Using summary statistics from 21 GWAS meta-analyses of phenotypes related to cardiovascular risk (Supplementary Table 2), the strongest association was between *PALMD* genetically-determined expression in the aortic valve and CAVS ($z = -7.2$, $P = 6.0 \times 10^{-13}$, Fig. 5 and Supplementary Data 14). We observed significant associations between genetically-determined expression of *PALMD* in several tissues (brain anterior cingulate cortex, esophagus muscularis, tibial nerve and subcutaneous adipose tissue) and atrial fibrillation ($z$ from 4.4 to 5.2, $P < 1 \times 10^{-5}$), with a strong probability of colocalization (PP4 from 61.4% to 97.0%) (Fig. 6 and Supplementary Fig. 4). Also, we found that predicted expression of *PALMD* in subcutaneous adipose tissue was associated with cardio-embolic stroke ($z = 3.7$, $P = 0.00024$), but the probability of colocalization was low (PP4 = 5.4%) (Fig. 6). Using the false-discovery rate threshold, there was a significant inverse association between *PALMD* genetically-determined expression in the aortic valve and ischemic stroke ($z = -3.1$, $P = 0.0019$), with a low probability of colocalization (PP4 = 15.7%) (Fig. 6).

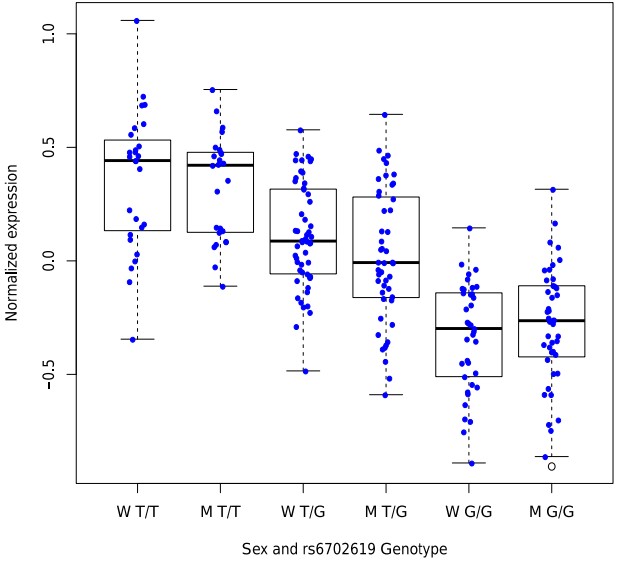

**Fig. 3 *PALMD* expression in 233 human aortic valves.** Boxplot representing *PALMD* expression according to sex and rs6702619 genotype in aortic valve tissue from 117 women and 116 men with CAVS. W: Women, M: Men. "G" is the CAVS risk allele at rs6702619. Whiskers represent the most extreme data points, which are no more than 1.5 times the interquartile range. The empty circle represents an outlier data point. Box boundaries represent the first and third quartiles. The center mark represents the median. $P_{het} = 0.36$ for the effect of rs6702619 on expression between sex (Woolf's test for heterogeneity).

## Discussion

A pheWAS approach on a wide range of health conditions showed that genetically-determined expression of *PALMD* in the aortic valve is specifically associated with aortic valve-related outcomes such as CAVS, aortic valve disorders and aortic valve replacement/repair. The association with CAVS was significantly stronger in women compared to men.

The mechanism responsible for the association between *PALMD* expression and CAVS remains presently unknown[10]. The protein coded by this gene, palmdelphin, has been shown to control myoblast differentiation by an unknown process and to

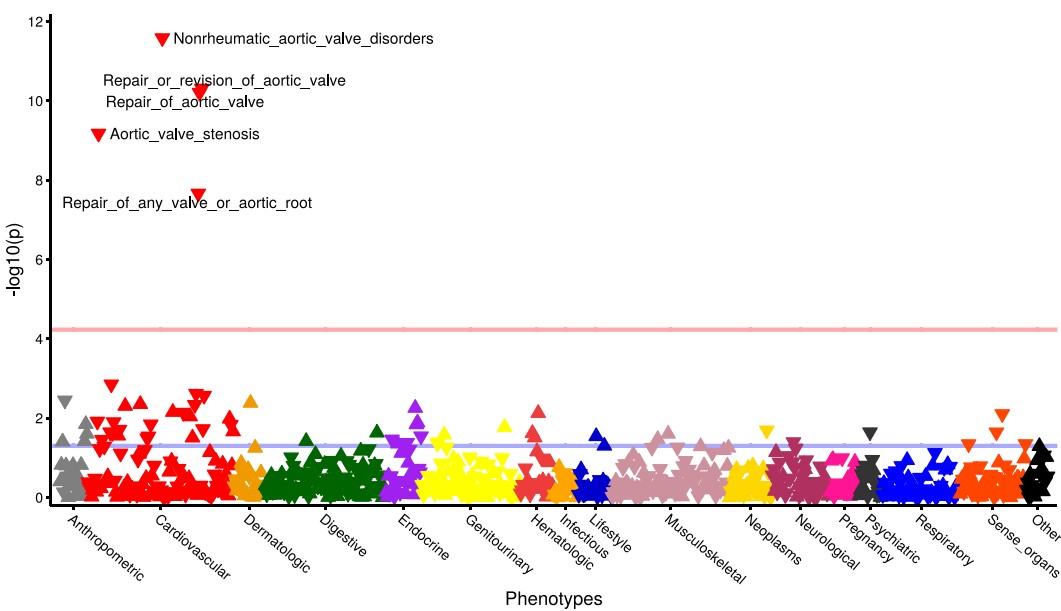

**Fig. 4 PheWAS of *PALMD* genetically-determined expression in the aortic valve.** Each triangle represents a different phenotype in the UK Biobank ($n = 852$). Triangles pointing up and down are positive and negative associations with *PALMD* genetically-determined expression in the aortic valve, respectively. The pink horizontal line represents the threshold for significance after correcting for multiple testing ($P = 0.05/852 = 5.9 \times 10^{-5}$). The blue horizontal line represents the threshold for nominal significance ($P = 0.05$).

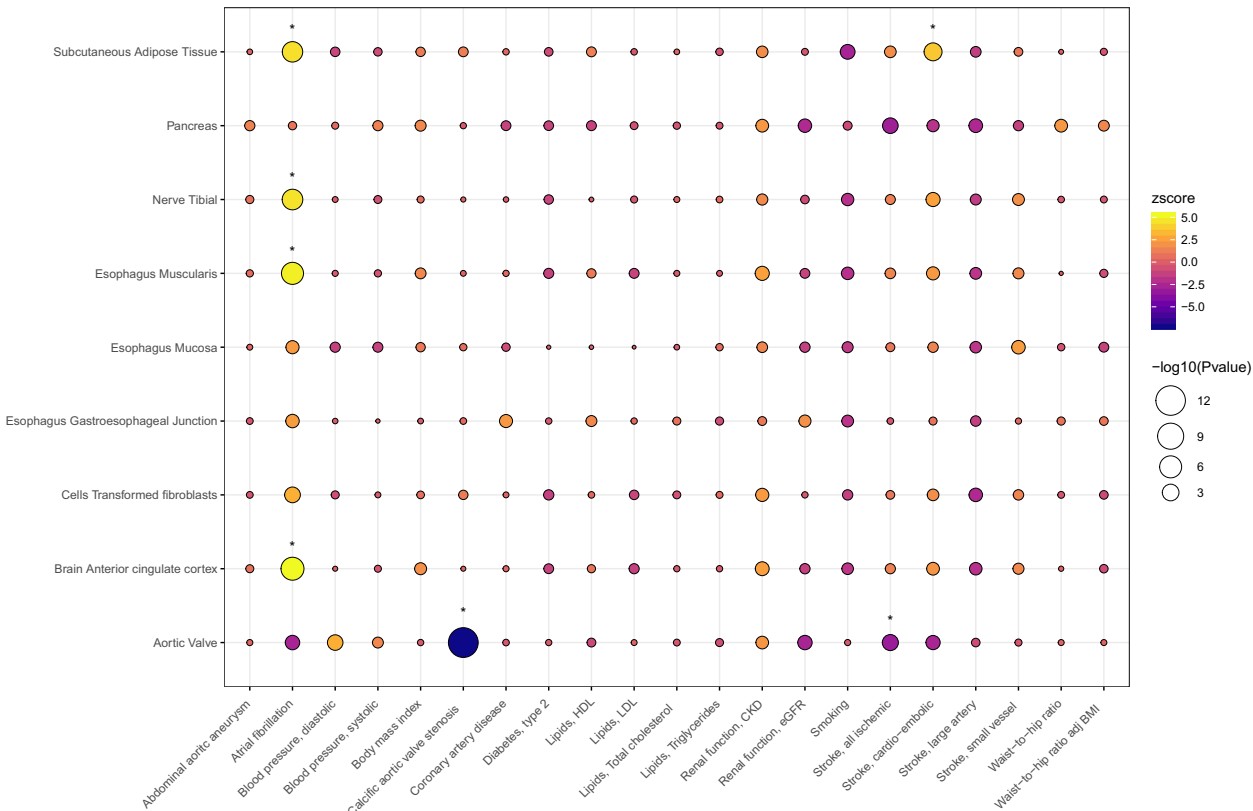

**Fig. 5 *PALMD* genetically-determined expression in 9 tissues and 21 cardiovascular phenotypes.** Analyses were performed using S-PrediXcan[16] and summary statistics from available GWAS meta-analyses (Supplementary Table 2). HDL high-density lipoprotein, LDL low-density lipoprotein, CKD chronic kidney disease, eGFR estimated glomerular filtration rate, BMI body-mass index. *$P_{FDR}$ < 0.05.

modulate the response to DNA damage in osteosarcoma cell lines[11,19]. These effects could contribute to the pathobiology of CAVS, which involves progressive fibrosis and mineralization of aortic leaflets[4].

Interestingly, although *PALMD* is expressed in various tissues, we found that the association with CAVS was restricted to genetically-determined expression in the aortic valve. This is concordant with our previous finding that CAVS risk alleles and increasing disease severity are both associated with decreased mRNA levels of *PALMD* in valve tissues[9]. In the eight tissues from the GTEx project for which a model could be developed to predict *PALMD* expression, we found no significant association with CAVS. Among 49 tissues, the lead CAVS risk variant, rs6702619, was only associated with *PALMD* expression in the aortic valve, without any significant association with other genes in cis or in trans. The underlying process explaining tissue-type specificity with CAVS despite a wide expression of *PALMD* is likely multifactorial and could involve tissue-specific regulation by transcription factors, tissue-restricted post-transcriptional processes, tissue-specific networks and redundancy of biological function by paralogues[20]. Our previous analysis has identified rs6702619, located 65 kb from the transcriptional start site of *PALMD*, as the most likely causal variant at this locus[21]. Loss-of-function coding variants in *PALMD* are rare (all frequencies <1 × $10^{-4}$). The ratio of observed to expected loss-of-function variants in gnomAD data v.2.1.1 is below 1 (0.41 [0.25–0.70]), suggesting some degree of intolerance[22].

The absence of a significant effect of predicted *PALMD* expression on a wide range of health conditions according to available data suggests that its modulation has likely a limited impact on other organs and systems. Among other phenotypes related to cardiovascular risk, there were modest associations

with atrial fibrillation (strongest for brain tissue, z = 5.2, $P = 2.2 \times 10^{-7}$) and cardio-embolic stroke (subcutaneous adipose tissue, z = 3.7, $P = 0.00024$) when looking at predicted expression in non-cardiac tissues. Notably, the signals showed a high probability of colocalization for atrial fibrillation but not for cardio-embolic stroke. The effects could potentially be underestimated, considering the lower variance explained in these tissues (<10%). These findings could indicate a theoretical risk of increasing *PALMD* expression globally although results should be replicated in independent studies.

The stronger effect of predicted *PALMD* expression in the aortic valve on CAVS observed in women compared to men is consistent with the results reported for the lead variant rs6702619 in a recent GWAS meta-analysis[21]. Whether this phenomenon is due to sex-specific pathophysiological mechanisms remains unknown and needs further exploration. Aortic valves from women with CAVS were shown to have more fibrosis and less calcification for a similar stenosis severity compared with men[23]. Women could therefore theoretically be more susceptible to a potential increase in valve fibrosis. The lead *PALMD* CAVS risk variant has also been associated with the presence of aortic stenosis in patients with a bicuspid aortic valve (BAV), a congenital anomaly more frequent in men[24] in which the aortic valve has two cusps instead of three, and with aortic root diameter in patients with a tricuspid valve[8]. We showed previously a similar effect on the risk of CAVS when restricting to patients with a tricuspid or a bicuspid valve[21], suggesting that the risk is not mediated through BAV.

This study has some limitations. The genetic component of *PALMD* expression was predicted using models trained on datasets including a modest number of individuals (between 102 and 328). A lower proportion of *PALMD* expression variance

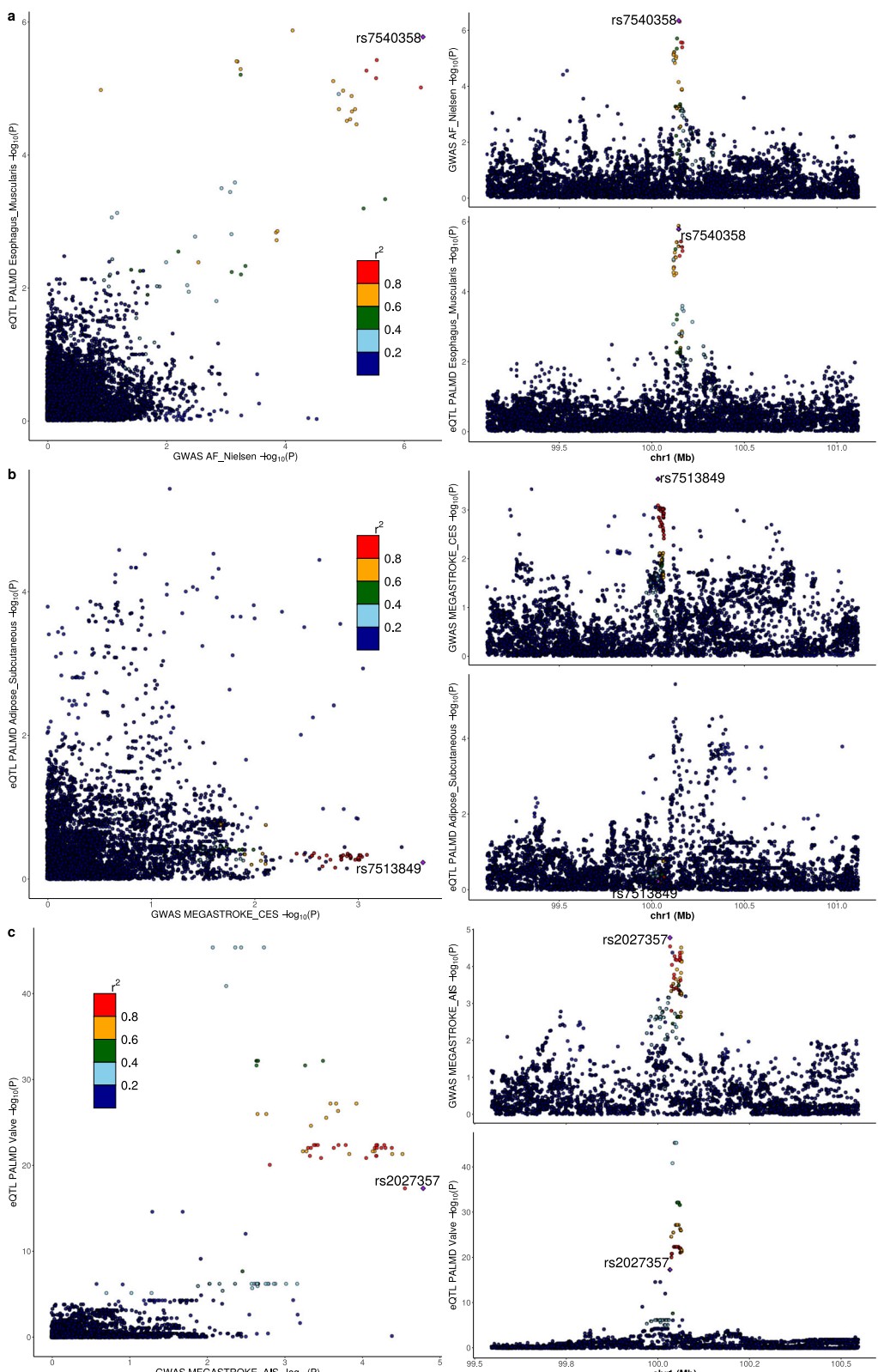

**Fig. 6 *PALMD* eQTL in selected tissues and GWAS association with atrial fibrillation and stroke.** LocusCompare plots[30] showing the relationship between *PALMD* eQTL and GWAS association with atrial fibrillation and stroke for variants located within 1 Mb of *PALMD*. **a** Esophagus muscularis and atrial fibrillation, colocalization PP4 = 97.0%. **b** Subcutaneous adipose tissue and cardio-embolic stroke, colocalization PP4 = 5.4%. **c** Aortic valve and all ischemic stroke, colocalization PP4 = 15.7%. GWAS associations were obtained from Nielsen et al.[31] and the MEGASTROKE Consortium[32]. The lead GWAS variant is annotated.

could be explained in other tissues compared to the aortic valve. Effects emerging from more important variations therefore cannot be excluded. We did not have genetic instruments to evaluate the effects in other cardiac and vascular tissues. The causal variant and the mechanism by which it affects *PALMD* expression and aortic valve stenosis remain to be validated in experimental models. Echocardiographic measurements were not available in UK Biobank. Whether PALMD or related pathways could be modulated for a future therapy is presently unknown and further research is needed to tease out the molecular processes underlying their impacts on the biology of the aortic valve. The approach used cannot predict off-target effects of a potential therapy targeting *PALMD*.

Taken together, these results suggest that *PALMD* is a promising molecular target for CAVS. There are no adverse effects predicted by a genetic increase of its expression in the aortic valve based on available data. Individuals with lower *PALMD* expression in the aortic valve, which can be predicted by their genotype, and women could potentially derive more benefits from a therapy increasing *PALMD* expression. Analyses in other tissues suggested a potential increase in risk of atrial fibrillation. Further studies are needed to confirm the biological mechanisms linking *PALMD* to CAVS and evaluate the safety and efficacy of agents modulating its expression.

## Methods

**Study cohorts**. The UK Biobank is a large prospective cohort of about 500,000 individuals between 40 and 69 years old recruited from 2006 to 2010 in several centers located in the United Kingdom[12]. Genome-wide genotypes including centrally imputed data are available for all participants[25]. Samples with call rate <95%, outlier heterozygosity rate, gender mismatch, non-white British ancestry, related samples (second degree or closer), samples with excess third-degree relatives (>10), or not used for relatedness calculation were excluded. CAVS diagnosis was determined using diagnoses and procedure codes as previously described[21]. Briefly, CAVS was defined as ICD10 code number I35.0 or I35.2. Participants with a history of rheumatic fever or rheumatic heart disease as determined by ICD10 codes I00–I02 and I05–I09 were excluded from the CAVS group. We included all other participants in the control group, except for those with OPCS-4 codes K26 (plastic repair of aortic valve) or K30.2 (revision of plastic repair of aortic valve) or a self-reported diagnosis of CAVS, which were excluded from the analysis. A total of 1350 individuals with CAVS and 349,043 individuals without CAVS were included. A total of 353,378 individuals (mean age 57 years, 54% female) were included in the pheWAS analyses. The present analyses were conducted under UK Biobank data application number 25205.

QUEBEC-CAVS is a case-control cohort study of patients undergoing cardiac surgery at the Institut universitaire de cardiologie et de pneumologie de Québec–Université Laval[9]. The cohort includes 2026 individuals (mean age 72 years, 64% male): 1009 individuals with nonrheumatic tricuspid CAVS undergoing aortic valve replacement and 1017 individuals without CAVS, most of which underwent a procedure for coronary artery disease. European ancestry was confirmed based on genotyping data. The study was approved by the IUCPQ-UL ethics committee and all patients signed an informed consent for the realization of genetic studies.

**Gene expression in the aortic valve and risk of CAVS**. We used our aortic valve eQTL dataset[9], derived from 233 individuals, to obtain the genetically-determined expression of *PALMD* in the aortic valve in the UK Biobank and QUEBEC-CAVS participants. We used the PredictDB pipeline to generate the prediction model[17]. Briefly, gene expression data was adjusted for age, sex, the first three principal components and 30 PEER factors[26]. Non-ambiguous variants (i.e., excluding A/T, G/C) with minor allele frequency ≥0.01 located within 1 Mb of the gene of interest were selected. An elastic net linear model was fit using the allelic dosages, with parameters alpha = 0.5 and lambda chosen by a 10 folds cross-validation. The weights generated with this procedure were used to predict gene expression from the individual genotypes in the UK Biobank and QUEBEC-CAVS using the PrediXcan software[17]. Logistic regression models were used to evaluate the relationship between predicted gene expression and CAVS in the UK Biobank and QUEBEC-CAVS, adjusting for age, sex, and the first 10 principal components. We then performed a fixed-effects inverse-variance weighted meta-analysis as implemented in the *rmeta* package in R.

**Sex-stratified analyses**. Sex-stratified analyses in the UK Biobank and QUEBEC-CAVS were performed to further characterize the association of genetically-determined expression of *PALMD* in the aortic valve with CAVS. *PALMD*

expression in 233 aortic valve tissues was compared between women (*n* = 117) and men (*n* = 116) using a Mann–Whitney test. Heterogeneity in gene expression according to the lead CAVS risk variant rs6702619 genotype was determined using Woolf's test for heterogeneity.

**PheWAS of genetically-determined expression in aortic valve**. To evaluate the effect of the *PALMD* genetically-determined expression in the aortic valve, we performed a phenome-wide association study (pheWAS) in 353,378 European individuals from the UK Biobank, as preformed previously[21]. Briefly, we selected 852 phenotypes, including anthropometric traits, health questionnaires, International Classification of Diseases 10th Revision (ICD10) diagnostic codes, Office of Population Censuses and Surveys Classification of Surgical Operations and Procedures, 4th Revision (OPCS-4) procedure codes, diagnostic procedure codes and laboratory markers (Supplementary Data 2). Diagnostic and procedure codes have been reviewed and curated to include both frequent specific conditions (e.g., myocardial infarction) and combinations of related diagnoses (e.g., ischemic heart diseases). Binary phenotypes (e.g., diagnostic codes) were analyzed using additive logistic regression models adjusting for age, sex and the first 10 ancestry-based principal components using SNPTEST v2.5.2[27]. Continuous variables were examined to exclude outlier values and quantile normalization was applied. Linear regression models were used with the covariates described above. Sex-specific phenotypes (e.g., cervical or prostate cancer) were analyzed in women or men only. Results were plotted using the *PheWAS* package in R. We performed the same analysis using the lead CAVS risk variant, rs6702619, as the dependent variable.

**Transcriptome-wide eQTL analysis of lead CAVS risk variant**. We performed eQTL analyses for rs6702619 in 48 tissues included in the GTEx project v7 (European samples), as well as in our aortic valve expression dataset using QTLtools v1.1[28]. All annotated genes (cis and trans) were evaluated. The following covariates were included: first three principal components, PEER factors according to the number of samples[29], sex, genotyping platform (GTEx only), and age (aortic valve only).

**Prediction of gene expression in other tissues**. We used the prediction models available in the PredictDB Data Repository for 48 tissues, which were derived from the European samples in GTEx project v7, to predict *PALMD* gene expression[17]. Only 8 non sex-specific tissues had weights available for the *PALMD* gene: brain anterior cingulate cortex, transformed fibroblasts, gastroesophageal junction, esophagus mucosa, esophagus muscularis, tibial nerve, pancreas, subcutaneous adipose tissue. In the PredictDB pipeline, a model is considered significant if the average Pearson correlation coefficient between predicted and observed expression during nested cross validation is greater than 0.1 (equivalent to $R^2 > 0.01$) and the estimated *p*-value for this statistic is less than 0.05[17].

**PheWAS of genetically-determined expression in other tissues**. We used the pheWAS procedure described above to evaluate the association between the predicted expression of *PALMD* in the 8 tissues and 852 phenotypes.

**Impact of expression on cardiovascular phenotypes**. We combined summary statistics from large GWAS meta-analyses (Supplementary Table 2) with the gene expression prediction models to determine the association between genetically-determined *PALMD* expression in the aortic valve, as well as the 8 other tissues with 21 phenotypes related to cardiovascular risk using S-PrediXcan[16].

**Statistical analysis**. Analyses were performed using R version 3.5.1 unless otherwise specified. To correct for multiple testing, *p*-values were adjusted according to false-discovery rate (Benjamini and Hochberg), Benjamini and Yekutieli and Bonferroni methods. The number of tests was set to 852 in the pheWAS for genetically-determined expression of *PALMD* in the aortic valve and rs6702619. In the pheWAS for genetically-determined expression of *PALMD* in other tissues, the number of tests was set to 6,816 (8*852). In the eQTL analyses, the number of tests was set to 42,052 for the aortic valve and 1,147,088 for the other tissues (total number of transcripts). In the S-PrediXcan analyses, the number of tests was set to 189 (9*21). An adjusted *p*-value <0.05 was considered significant. The *LocusCompareR* package in R was used to generate LocusCompare plots[30] to visualize significant associations. The *coloc* package in R was used to perform colocalization analyses for significant associations. Variants located at +/−500 kb of the lead GWAS variant were selected and a posterior probability for a shared causal variant (PP4) above 60% was considered as indicative of colocalization.

**Reporting summary**. Further information on research design is available in the Nature Research Reporting Summary linked to this article.

## Data availability
The microarray gene expression data set on human aortic valves was deposited in Gene Expression Omnibus with accession number GSE102249. UK Biobank data are available following an established application process. GTEx project data are available

at https://www.gtexportal.org/home/ and on dbGaP (accession phs000424.v8.p2). PredictDB transcriptome prediction models are available at http://predictdb.org/. Links for the publicly available summary statistics used in this work are available in Supplementary Table 2.

## Code availability

The software used in this analysis is publicly available at the URLs below: PredictDB Pipeline: https://github.com/hakyimlab/PredictDB_Pipeline_GTEx_v7; PrediXcan and S-PrediXcan: https://github.com/hakyimlab/MetaXcan; rmeta: https://cran.r-project.org/web/packages/rmeta/index.html; SNPTEST: https://mathgen.stats.ox.ac.uk/genetics_software/snptest/snptest.html; PheWAS: https://github.com/PheWAS/PheWAS; QTLtools: https://qtltools.github.io/qtltools/; LocusCompareR: https://github.com/boxiangliu/locuscomparer; coloc: https://cran.r-project.org/web/packages/coloc/index.html.

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

## Acknowledgements

We thank the research team and the participants who provided samples to the biobank of the Institut universitaire de cardiologie et de pneumologie de Québec-Université Laval (IUCPQ-UL). This research has been conducted using the UK Biobank Resource (data application number 25205). B.J.A. holds a Junior 2 Research Scholar award from the FRQS. P.M. holds a Fonds de Recherche du Québec-Santé (FRQS) Research Chair on the Pathobiology of Calcific Aortic Valve Disease. Y.B. holds a Canada Research Chair in Genomics of Heart and Lung Diseases. S.T. holds a Junior 1 Clinical Research Scholar award from the Fonds de Recherche du Québec-Santé (FRQS). This work was supported by a grant from the Canadian Institutes of Health Research (PJT–162344) to S.T.

## Author contributions

P.M., Y.B., and S.T. contributed to the conception and study design. N.G., P.M., Y.B., and S.T. contributed to data collection. Z.L. and S.T. contributed to data analysis. B.J.A., P.M., Y.B., and S.T. contributed to data interpretation. Z.L. and S.T. drafted the manuscript. Z.L., N.G., B.J.A., P.M., Y.B., and S.T. revised the manuscript.

## Competing interests

The authors declare no competing interests.
