## [Peer Review File · Communications Biology]

Reviewers' comments:

Reviewer #1 (Remarks to the Author):

In this paper, Li et al. show that genetically determined gene expression of PALMD associates with CAVS in two cohorts. They find that genetically determined gene expression in tissues other than aortic valve associates with other cardiovascular phenotypes. They also test the phenomewide effects of variation in PALMD expression, and explore the tissue specificity of these associations. These studies are performed in the context of assessing the phenomewide effects of PALMD modulation as a therapeutic target for CAVS. In this context, it is also important to know which associations are absent, which the authors also discuss.

General comments:

1. It is ideal to have some replication for the positive and important negative pheWAS associations observed in the UK Biobank. In order to achieve this, the authors could either use a strategy to split the data for pheWAS, similar to ones performed previously (Rao, AS., et al. "Large-scale phenomewide association study of PCSK9 variants demonstrates protection against ischemic stroke." *Circulation: Genomic and Precision Medicine* 11.7 (2018): e002162), or seek to replicate important associations that they discuss in QUEBEC-CAVS if the relevant phenotypes are available.
2. How is CAVS defined across both cohorts used? Please include the specific definition used in both cases in this publication. The main results show a significant association with aortic valve related phenotypes. Were these codes used in the definition of CAVS in the UK Biobank GWAS? If they were, the associations of the variants in the PrediXcan model with these phenotypes might be expected to some extent based on previous results (acknowledging that the associations reported in this paper are between phenotypes and effectively linear combinations of variants).
3. Have the authors considered performing either finemapping or colocalization analysis using methods such as COLOC (Giambartolomei, Claudia, et al. "A Bayesian framework for multiple trait colocalization from summary association statistics." *Bioinformatics* 34.15 (2018): 2538-2545) or eCAVIAR (Hormozdiari, Farhad, et al. "Colocalization of GWAS and eQTL signals detects target genes." *The American Journal of Human Genetics* 99.6 (2016): 1245-1260) to determine the likely causal variant in the locus? Considering that this study uses genetically predicted gene expression, this information can be a useful addition to the discussion and will provide further insight into cis-regulation of PALMD expression. It would also further help determine if the lead variant used by the authors, rs6702619, seems to be the variant that drives both CAVS and PALMD expression association signals.
4. The model used in the aortic valve includes 17 variants to predict gene expression. The authors could consider performing a conditional analysis on the PALMD expression signal in aortic valve to determine if there might be multiple independent effects. Colocalization results with CAVS across these independent effects (if any) would be a valuable addition. The authors can also use a LocusCompare (locuscompare.com, Liu, Boxiang, et al. "Abundant associations with gene expression complicate GWAS follow-up." *Nature genetics* 51.5 (2019): 768-769) style plot to observe this relationship.
5. While the authors mention that the variants associated with CAVS are not associated with gene expression in 48 other tissues, cis-genetics still seems to be able to predict gene expression in other tissues. This could also suggest the presence of independent cis-associations with PALMD expression, and the locuscompare plot suggested above could be extended to eQTL associations across various tissues.
6. Have the authors released some or all of the aortic valve gene expression / eQTL dataset, or do they plan to? The data set is unique and could be valuable to the broader research community. There are also platforms like the aforementioned LocusCompare website that add value by integrating the

eQTL results with a wide variety of phenotypes.

Minor points:

1. The authors seek to identify the effects of variation in PALMD expression on phenotypes based on an initial discovery of the relationship between PALMD expression and CAVS in another study. Coding variants could also have a large impact on gene function, as some other studies focused on potential therapeutic target genes have done. Are there coding variants that could also be used to test this association? If not (or are present at very low frequencies), is PALMD generally loss-of-function intolerant? It will help to include some detail on the genetic variation observed in PALMD in the discussion.
2. In the introduction, the authors mention that the variants associated with CAVS do not affect PALMD expression in 48 other tissues from the GTEx dataset. Are there subthreshold signals that might not reach significance due to small sample sizes in some tissues? It would help to add in min p-value of association across other tissues and/or a supplementary figure highlighting tissue specificity of association.
3. Further, while the relationship between PALMD and CAVS has been shown in a relevant tissue and is presented as likely causal in a previous publication, do the variants associated with PALMD expression show an association with any other genes either in cis- or trans- across tissues?
4. In the introduction, the authors state that pheWAS allows the prediction of the beneficial as well as the adverse effects of modification of a specific molecular target. However, this does not include the effects that could result from off-target effects of the molecule used to achieve this modification. It would help to add a statement expressing this limitation.
5. In the Discussion section, the authors only very briefly state that the mechanisms linking PALMD to CAVS is unknown, and include a single sentence on the biological relevance of the protein palmdelphin. It would help the reader to have a few more sentences on the general pathophysiology of CAVS.
6. In the discussion, the authors should include effect size and p-value information where modest associations are described.
7. Genetically-determined expression of PALMD seems to associate with smoking behavior across multiple tissues, but this effect is not seen in aortic valve. This might be interesting to add to the discussion in more detail. Do the authors think there is a biological interpretation for this pattern or for this association in general, or could it be an artifact of lower power?
8. What criteria specifically regarding relatedness/ethnic background were used to determine the study cohort? Please include this information to benefit replication, especially since individual IDs tend to vary across UK Biobank applications.
9. While there do not seem to be pheWAS association signals that are very close to the significance threshold, using Bonferroni correction across 834 phenotypes might be conservative depending on the correlation between phenotypes used. The authors could consider also reporting results using an alternative multiple testing strategy that is more liberal but sensitive to correlation (eg. Benjamini-Yekutieli).

Reviewer #2 (Remarks to the Author):

Reviewers comments on "A phenome-wide approach establishes a specific association between the expression of PALMD in the aortic valve and calcific aortic valve stenosis" by Li et al.

The paper is focused on the PALMD-locus which has previously been established in GWAS/TWAS of CAVS (PMID: 29511194, 29511167). The authors use a pheWAS approach to assess the potential

therapeutic utility and adverse effects of modulating PALMD expression in the aortic valve or globally. The authors conclude that the inverse association between PALMD expression and CAVS risk is specific to the aortic valve tissue and that this association is stronger in women compared to men and both of these claims are consistent with results of previous studies by the authors themselves (PMID: 29511167, 32141789). The major novel claims of the current paper are that no adverse effects are predicted by a genetic increase of PALMD expression in the aortic valve but a positive association between predicted PALMD expression and atrial fibrillation/cardioembolic stroke was observed in other tissues.

My main concern regards the claims of lack of effects on other phenotypes than CAVS. In my opinion these claims would be more convincing if the analysis was more robust and comprehensive. Several additions or adjustments could be made. Most importantly this applies to the selection of phenotypes (comment #1), which is an essential part of this paper. Otherwise I think that the wording needs to be toned down. This is further explained in my specific comments and suggestions below.

On the whole, the paper is clearly written and seems to use sound methodology. It addresses an interesting and relevant issue, as it attempts to provide knowledge that helps in translating GWAS/TWAS results to clinical significance.

Specific comments and suggestions:

1. Selection of phenotypes for the UK Biobank pheWAS: Too many of the phenotypes chosen are very unspecific (e.g. "heart cardiac problem") and many of them overlap (e.g. the same disorder according to ICD, "NCI Clinical" and OPCS) while specific ICD codes for very relevant conditions are missing (e.g. aortic valve insufficiency, mitral valve prolapse, hypertrophic/dilated cardiomyopathy, atrial fibrillation ect. ect.)

In many cases the specific diagnoses would likely be more useful than the unspecific/general diagnoses. For example, atrial fibrillation associates with predicted PALMD expression in brain and esophagus in the S-PrediXcan analysis (ST14) but the phenotype "heart_arrhythmia" is flat in the UKB-pheWAS in both tissues. Furthermore, since CAVS variants at the PALMD locus have been associated with aortic root size, BAV and cardiac septal defects (PMID: 29511194) it would have been very relevant to include congenital heart defects and echocardiographic traits.

I realize that in many cases these phenotypes are simply not available, but UK Biobank has a large atrial fibrillation sample set, why is that not included?

2. Correcting for multiple testing: It could be argued that a conventional Bonferroni correction is too conservative in the case of the UKB-pheWAS, in particular since many of the phenotypes overlap (as pointed out in #1) and are likely highly correlated. Would a less stringent approach, such as false discovery rate procedure be more appropriate? This is especially important when the main conclusion is about lack of association.

3. Information on prediction models (ST5):

a. The calculations of predicted PALMD expression could be made more transparent to the reader. It would be very informing to list the variants underlying the calculations and their assigned weights, f.ex. in supplementary table 5.

b. The authors mention the low proportion of variance in expression (2-20%) explained in other tissues than the aortic valve as one of the limitations of the study. For a reader that is not knowledgeable about the details of PrediXcan (such as myself) this raises the question of what is considered high and low? Is there any consensus on what is an acceptable proportion for a particular model to be used to test for trait associations? Scanning the literature does not provide easy answers, if they exist it might be useful to provide the reader with some insight and a reference.

4. **A pheWAS using the lead CAVS variant at PALMD?** This would likely be simple to perform in the same way as done for the predicted expression of PALMD (and as done for lead variants at three other CAVS loci here: PMID: 32141789). As the authors mention in the introduction, lead variants at the locus affect PALMD expression in the aortic valve and not in 48 other tissues in GTEx (although it says 44 tissues in the cited paper: PMID: 29511167, is this a mistake or has the analysis been repeated with additional tissues?). It would thus be interesting to know if a pheWAS using the lead variant is consistent with the pheWAS using the predicted expression score in aortic tissue.

5. **Wording of conclusions:** All in all, there are several limitations to the study, most of which the authors mention in the discussion section. Some improvements may be possible (as suggested in #1-4) but other shortcomings seem harder to avoid, most notably the lack of analysis in other cardiac and vascular tissues than the aortic valve. It is important to note, that despite the shortcomings the analysis is informing and based on quality data, the aortic root sample set and the UK Biobank extensive phenotypic dataset. However, it is my opinion that in light of the limitations, the wording in the discussion section is a bit strong, for example: *"The absence of a significant effect of predicted PALMD expression on a wide range of health conditions suggests that its modulation has likely a limited impact on other organs and systems."* and *"There are no adverse effects predicted by a genetic increase of its expression in the aortic valve."* I would suggest toning this down, or for example adding phrases like "according to data available to us". Also the *"modest associations with atrial fibrillation and cardio-embolic stroke"* might be underestimated in light of the before mentioned limitations (i.e. no analysis in heart tissue, only 2-20% of variance in expression explained by models). Finally, I think it is worth mentioning as a limitation that the phenotype list is not complete, and that f.ex. phenotypes that have been associated with CAVS-variants at PALMD (PMID: 29511194, PMID: 28394258: ASD/VSD/echocardiographic measurements, see #1) were not available for analysis.

6. **Sex-stratified analysis:** Sex-stratified analysis was only performed for the association of genetically-determined expression of PALMD in the aortic valve with CAVS. It would be interesting to also perform sex-stratified analysis for the AF/stroke associations in other tissues if possible.

7. **Improved phenotypic description:** A description of the terms "clinical - NCI" and "clinical" should be included in the methods section and/or table footnotes.

8. **Comment on the discussion about potential mechanisms responsible for the association between PALMD expression and CAVS:** In the discussion section it says: *"The lead PALMD CAVS risk variant has also been associated with the presence of aortic stenosis in patients with a bicuspid valve, a congenital anomaly in which the aortic valve has two cusps instead of three."* However, this condition is significantly more frequent in men and therefore cannot explain the stronger association between genetically-determined PALMD expression and CAVS observed in women.

First, I do not agree with the above statement that just because BAV in general is more in common in men, PALMD expression levels could not be mediating a stronger risk of BAV (and therefore CAVS) in women. However, it might be worth discussing in this context that the aortic valve eQTL dataset is only based on individuals with tricuspid valves which is an argument against the notion that PALMD expression mediates risk of CAVS only by increasing risk of BAV. Furthermore, it might also be worth mentioning that one of the lead CAVS associated variants at PALMD has been associated with increased aortic root dimension (PMID: 28394258) and that this association remains after removing BAV cases from the analysis (PMID: 29511194).

9. **Comment on the following statement in the discussion:** *"Interestingly, although*

PALMD is expressed in various tissues, we found that the association with CAVS was restricted to genetically-determined expression in the aortic valve. Accordingly, in the eight tissues from the GTEx project for which a model could be developed to predict PALMD expression, we found no significant association with CAVS." - it would be appropriate to mention that this complies with the authors previous study reporting that "The CAVS risk alleles and increasing disease severity are both associated with decreased mRNA expression levels of PALMD in valve tissues." and that ..."
rs6702619-PALMD association was not significant in 44 tissues from GTEx, suggesting that the change in expression is specific to aortic valve tissue"....and cite that paper at this point in the discussion (PMID: 29511167).

Response to reviewers COMMSBIO-20-0584

We thank the reviewers for their thorough assessment and constructive criticism. We updated the manuscript accordingly and believe it is now significantly improved. Please find below the point-by-point answers.

Reviewer #1:

In this paper, Li et al. show that genetically determined gene expression of PALMD associates with CAVS in two cohorts. They find that genetically determined gene expression in tissues other than aortic valve associates with other cardiovascular phenotypes. They also test the phenomewide effects of variation in PALMD expression, and explore the tissue specificity of these associations. These studies are performed in the context of assessing the phenomewide effects of PALMD modulation as a therapeutic target for CAVS. In this context, it is also important to know which associations are absent, which the authors also discuss.

General comments:

1. It is ideal to have some replication for the positive and important negative pheWAS associations observed in the UK Biobank. In order to achieve this, the authors could either use a strategy to split the data for pheWAS, similar to ones performed previously (Rao, AS., et al. "Large-scale phenome-wide association study of PCSK9 variants demonstrates protection against ischemic stroke." *Circulation: Genomic and Precision Medicine* 11.7 (2018): e002162), or seek to replicate important associations that they discuss in QUEBEC-CAVS if the relevant phenotypes are available.

The numbers of phenotypes and samples available in QUEBEC-CAVS are not sufficient to perform a pheWAS analysis. The analyses with S-PrediXcan using summary statistics from several consortia provide other sources of data for relevant cardiovascular phenotypes to support the associations observed in the pheWAS. In that regard, we added smoking status as another variable in this section (see also answer to Minor point #7 below). As for the association with CAVS, results are consistent in the UK Biobank and QUEBEC-CAVS (Figure 1). We did not split the data in UK Biobank to avoid a loss in power, which would make it more difficult to identify or exclude associations with rare phenotypes.

We added the following sentence in the Discussion section (regarding atrial fibrillation and stroke):

These findings could indicate a theoretical risk of increasing PALMD expression globally, although results should be replicated in independent studies.

2. How is CAVS defined across both cohorts used? Please include the specific definition used in both cases in this publication. The main results show a significant association with aortic valve related phenotypes. Were these codes used in the definition of CAVS in the UK Biobank

GWAS? If they were, the associations of the variants in the PrediXcan model with these phenotypes might be expected to some extent based on previous results (acknowledging that the associations reported in this paper are between phenotypes and effectively linear combinations of variants).

We added the definition of CAVS in each cohort in the Methods section. The definition in UK Biobank represents a combination of codes for the cases and control groups whereas the pheWAS were based on isolated codes. The association with CAVS was indeed expected based on our previous publications (see answer to Major point #3 below).

We added the following in the Methods section (for UK Biobank and QUEBEC-CAVS, respectively):

Briefly, CAVS was defined as ICD10 code number I35.0 or I35.2. Participants with a history of rheumatic fever or rheumatic heart disease as determined by ICD10 codes I00–I02 and I05–I09 were excluded from the CAVS group. We included all other participants in the control group, except for those with OPCS-4 codes K26 (plastic repair of aortic valve) or K30.2 (revision of plastic repair of aortic valve) or a self-reported diagnosis of CAVS, which were excluded from the analysis.

The cohort includes 1,009 individuals with nonrheumatic tricuspid CAVS undergoing aortic valve replacement and 1,017 individuals without CAVS, most of which underwent a procedure for coronary artery disease.

3. Have the authors considered performing either finemapping or colocalization analysis using methods such as COLOC (Giambartolomei, Claudia, et al. "A Bayesian framework for multiple trait colocalization from summary association statistics." *Bioinformatics* 34.15 (2018): 2538-2545) or eCAVIAR (Hormozdiari, Farhad, et al. "Colocalization of GWAS and eQTL signals detects target genes." *The American Journal of Human Genetics* 99.6 (2016): 1245-1260) to determine the likely causal variant in the locus? Considering that this study uses genetically predicted gene expression, this information can be a useful addition to the discussion and will provide further insight into cis-regulation of PALMD expression. It would also further help determine if the lead variant used by the authors, rs6702619, seems to be the variant that drives both CAVS and PALMD expression association signals.

Some of the proposed analyses have already been performed and published in our previous manuscripts^{1,2}. For example, PAINTOR analyses have been used to identify rs6702619 as the most likely causal variant at this locus.

We added a sentence in the Discussion to emphasize these previous findings:

Our previous analysis has identified rs6702619, located 65kb from the transcriptional start site of PALMD, as the most likely causal variant at this locus¹.

4. The model used in the aortic valve includes 17 variants to predict gene expression. The authors could consider performing a conditional analysis on the PALMD expression signal in aortic valve to determine if there might be multiple independent effects . Colocalization results with CAVS across these independent effects (if any) would be a valuable addition. The authors can also use a LocusCompare (locuscompare.com, Liu, Boxiang, et al. "Abundant associations with gene expression complicate GWAS follow-up." Nature genetics 51.5 (2019): 768-769) style plot to observe this relationship.

The variants with a strong effect on *PALMD* expression in the model are located nearby the lead variant rs6702619. We added a table with the list of variants and their respective weight (Supplementary Table 3). We also added a LocusCompare plot and colocalization analysis to illustrate the relationship between the effect of the variants on *PALMD* expression and CAVS (Figure 2, see Appendix).

We added the following in the Methods section:

The locuscomparer package in R was used to generate LocusCompare plots³ to visualize significant associations. The coloc package in R was used to perform colocalization analyses for significant associations. Variants located at +/- 500kb of the lead GWAS variant were selected and a posterior probability for a shared causal variant (PP4) above 60% was considered as indicative of colocalization.

We added the following sentence in the Results section:

There was a strong probability of colocalization (PP4=99.7%).

5. While the authors mention that the variants associated with CAVS are not associated with gene expression in 48 other tissues, cis-genetics still seems to be able to predict gene expression in other tissues. This could also suggest the presence of independent cis- associations with PALMD expression, and the locuscompare plot suggested above could be extended to eQTL associations across various tissues .

We added LocusCompare plots to illustrate the relationship between the effect of the variants on *PALMD* expression in the 9 tissues and the phenotypes for which significant associations were found (Figure 5, see Appendix and Supplementary Figures 4-5).

6. Have the authors released some or all of the aortic valve gene expression / eQTL dataset, or do they plan to? The data set is unique and could be valuable to the broader research community. There are also platforms like the aforementioned LocusCompare website that add value by integrating the eQTL results with a wide variety of phenotypes.

The microarray gene expression data set on human aortic valves was deposited in Gene Expression Omnibus with accession number GSE102249. The valve cis-eQTL data are available in our previous publication² as Supplementary Data.

Minor points:

1. The authors seek to identify the effects of variation in PALMD expression on phenotypes based on an initial discovery of the relationship between PALMD expression and CAVS in another study. Coding variants could also have a large impact on gene function, as some other studies focused on potential therapeutic target genes have done. Are there coding variants that could also be used to test this association? If not (or are present at very low frequencies), is PALMD generally loss-of-function intolerant? It will help to include some detail on the genetic variation observed in PALMD in the discussion.

According to gnomAD v.2.1.1 (including 141,456 samples), loss-of-function coding variants in *PALMD* are rare (all frequencies $<1 \times 10^{-4}$). Missense variants also have low frequencies (<0.01) and there is insufficient evidence to identify with certainty the ones that could be disruptive. Selecting variants with an impact on gene expression therefore appears to be the most logical approach. *PALMD* cannot be classified as either loss-of-function intolerant or tolerant based on the gnomAD data. Although the probability of being loss-of-function intolerant (pLI) is calculated at 0, the ratio of observed to expected loss-of-function variants is below 1 (0.41 [0.25 - 0.70]), suggesting some degree of intolerance.

We added the following in the Discussion section:

Loss-of-function coding variants in *PALMD* are rare (all frequencies $<1 \times 10^{-4}$). The ratio of observed to expected loss-of-function variants in gnomAD data v.2.1.1 is below 1 (0.41 [0.25 - 0.70]), suggesting some degree of intolerance⁴.

2. In the introduction, the authors mention that the variants associated with CAVS do not affect PALMD expression in 48 other tissues from the GTEx dataset. Are there subthreshold signals that might not reach significance due to small sample sizes in some tissues? It would help to add in min p-value of association across other tissues and/or a supplementary figure highlighting tissue specificity of association.

No association between the lead CAVS variant, rs6702619, and *PALMD* expression in the 48 tissues from the GTEx dataset had a p-value below 0.01 (minimum p-value was 0.013). The sample size (European samples) per tissue was between 70 and 359.

We corrected the sentence in the Introduction section to clarify (see also answer to Reviewer 2 point #4).

Of note, the variants associated with CAVS did not affect *PALMD* expression in 44 other tissues in the Genotype-Tissue Expression (GTEx) project.

All cis and trans eQTL with a p-value below 0.001 are now available in Supplementary Table 8 (see also answer to Minor point #3 below).

3. Further, while the relationship between PALMD and CAVS has been shown in a relevant tissue and is presented as likely causal in a previous publication, do the variants associated with PALMD expression show an association with any other genes either in cis- or trans - across tissues?

We obtained all the cis and trans eQTL associations between the lead CAVS risk variant, rs6702619, and gene expression in the 48 tissues in the GTEx v7 dataset (European samples) and the aortic valve. The association with *PALMD* expression in the aortic valve ($P=4.0E-46$) is by far the strongest observed. The next association in significance has a p-value of $1.1E-06$, which is not significant following Bonferroni or false-discovery rate correction.

We now show the cis and trans eQTL with a p-value below 0.001 in Supplementary Table 8).

We added the following in the Methods section:

Transcriptome-wide eQTL analysis of the lead CAVS risk variant

We performed eQTL analyses for rs6702619 in 48 tissues included in the GTEx project v7 (European samples) as well as in our aortic valve expression dataset using QTLtools v1.1⁵. All annotated genes (cis and trans) were evaluated. The following covariates were included: first three principal components, PEER factors according to the number of samples⁶, sex, genotyping platform (GTEx only) and age (aortic valve only).

We added the following sentences in the Results and Discussion sections:

Transcriptome-wide eQTL analyses for rs6702619 in 48 tissues included in the GTEx project v7 European samples as well as in our aortic valve expression dataset confirmed the known signal for *PALMD* expression in the aortic valve. No other association was significant following correction for multiple testing (Supplementary Table 8).

Among 49 tissues, the lead CAVS risk variant, rs6702619, was only associated with *PALMD* expression in the aortic valve, without any significant association with other genes in cis or in trans.

4. In the introduction, the authors state that pheWAS allows the prediction of the beneficial as well as the adverse effects of modification of a specific molecular target. However, this does not include the effects that could result from off-target effects of the molecule used to achieve this modification. It would help to add a statement expressing this limitation.

We agree with the reviewer.

We added the following sentence in the limitation section:

The approach used cannot predict off-target effects of a potential therapy targeting *PALMD*.

5. In the Discussion section, the authors only very briefly state that the mechanisms linking PALMD to CAVS is unknown, and include a single sentence on the biological relevance of the

protein palmdelphin. It would help the reader to have a few more sentences on the general pathophysiology of CAVS.

The precise mechanism linking *PALMD* to CAVS is currently under investigation. Nevertheless, some evidence regarding the function of palmdelphin points to effects on myoblast differentiation and fibrosis, which are processes involved in the pathophysiology of CAVS.

We added the following sentence in the Discussion section:

The protein coded by this gene, palmdelphin, has been shown to control myoblast differentiation by an unknown process and to modulate the response to DNA damage in osteosarcoma cell lines. These effects could contribute to the pathobiology of CAVS, which involves progressive fibrosis and mineralization of aortic leaflets⁷.

6. In the discussion, the authors should include effect size and p-value information where modest associations are described.

As suggested by the reviewer, we added the effect size and p-value when mentioning the associations in the Discussion:

Among other phenotypes related to cardiovascular risk, there were modest associations with atrial fibrillation (strongest for brain tissue, $z=5.2$, $P=2.2 \times 10^{-7}$) and cardio-embolic stroke (subcutaneous adipose tissue, $z=4.4$, $P=0.00024$) when looking at predicted expression in non-cardiac tissues.

7. Genetically-determined expression of *PALMD* seems to associate with smoking behavior across multiple tissues, but this effect is not seen in aortic valve. This might be interesting to add to the discussion in more detail. Do the authors think there is a biological interpretation for this pattern or for this association in general, or could it be an artifact of lower power ?

To further explore this potential association, we added smoking in the list of cardiovascular phenotypes in the S-PrediXcan analysis. Summary statistics was taken from a meta-analysis of UK Biobank and the TAG consortium⁸. We did not find evidence of a significant association (Figure 4, see Appendix and Supplementary Table 17). There was also no evidence of colocalization. Even when adjusting the p-values in the pheWAS using less stringent approaches than Bonferroni (FDR, Benjamini-Yekutieli), the association with smoking was not significant. Therefore, we conclude that there is not enough evidence to support an association with smoking.

8. What criteria specifically regarding relatedness/ethnic background were used to determine the study cohort? Please include this information to benefit replication, especially since individual IDs tend to vary across UK Biobank applications.

Only individuals of European ancestry were selected in the UK Biobank, QUEBEC-CAVS and GTEx datasets.

We added this information in the Methods section for UK Biobank and QUEBEC-CAVS:

Samples with call rate <95%, outlier heterozygosity rate, gender mismatch, non-white British ancestry, related samples (second degree or closer), samples with excess third-degree relatives (>10), or not used for relatedness calculation were excluded.

European ancestry was confirmed based on genotyping data.

9. While there do not seem to be pheWAS association signals that are very close to the significance threshold, using Bonferroni correction across 834 phenotypes might be conservative depending on the correlation between phenotypes used. The authors could consider also reporting results using an alternative multiple testing strategy that is more liberal but sensitive to correlation (eg. Benjamini-Yekutieli).

We now report the results with p-values adjusted for false-discovery rate and Benjamini-Yekutieli for all analyses (see Supplementary Tables 6,7,9-17). The use of less stringent corrections for multiple hypothesis testing did not change the interpretation of the pheWAS results (i.e. no phenotype reached statistical significance except for aortic valve related phenotypes for the genetically-determined expression in the aortic valve). The use of false-discovery rate identified a potential negative association between *PALMD* predicted expression in the aortic valve and ischemic stroke in the S-PrediXcan analyses (same direction of effect as CAVS).

We added the following in the Methods section:

*To correct for multiple testing, p-values were adjusted according to false-discovery rate (Benjamini & Hochberg), Benjamini & Yekutieli and Bonferroni methods. The number of tests was set to 852 in the pheWAS for genetically-determined expression of *PALMD* in the aortic valve and rs6702619. In the pheWAS for genetically-determined expression of *PALMD* in other tissues, the number of tests was set to 6,816 (8*852). In the eQTL analyses, the number of tests was set to 42,052 for the aortic valve and 1,147,088 for the other tissues (total number of transcripts). In the S-PrediXcan analyses, the number of tests was set to 189 (9*21). An adjusted p-value <0.05 was considered significant.*

We added the following in the Results and Discussion sections:

*Using the false-discovery rate threshold, there was a significant association between *PALMD* genetically-determined expression in the aortic valve and ischemic stroke ($z=-3.1$, $P=0.0019$), with a low probability of colocalization ($PP4=15.7\%$) (Figure 5).*

*In addition to CAVS, a potential beneficial effect of increasing *PALMD* expression in the aortic valve was identified for ischemic stroke ($z=-3.1$, $P=0.0019$).*

Reviewer #2:

The paper is focused on the PALMD-locus which has previously been established in GWAS/TWAS of CAVS (PMID: 29511194, 29511167). The authors use a pheWAS approach to assess the potential therapeutic utility and adverse effects of modulating PALMD expression in the aortic valve or globally. The authors conclude that the inverse association between PALMD expression and CAVS risk is specific to the aortic valve tissue and that this association is stronger in women compared to men and both of these claims are consistent with results of previous studies by the authors themselves (PMID: 29511167, 32141789). The major novel claims of the current paper are that no adverse effects are predicted by a genetic increase of PALMD expression in the aortic valve but a positive association between predicted PALMD expression and atrial fibrillation/cardioembolic stroke was observed in other tissues.

My main concern regards the claims of lack of effects on other phenotypes than CAVS. In my opinion these claims would be more convincing if the analysis was more robust and comprehensive. Several additions or adjustments could be made. Most importantly this applies to the selection of phenotypes (comment #1), which is an essential part of this paper. Otherwise I think that the wording needs to be toned down. This is further explained in my specific comments and suggestions below.

On the whole, the paper is clearly written and seems to use sound methodology. It addresses an interesting and relevant issue, as it attempts to provide knowledge that helps in translating GWAS/TWAS results to clinical significance.

We thank the reviewer for these thoughtful remarks. We addressed the points regarding the selection of phenotypes, changed some of the wording and performed additional analyses to add robustness to the results in response to the specific comments below.

Specific comments and suggestions:

1. Selection of phenotypes for the UK Biobank pheWAS: Too many of the phenotypes chosen are very unspecific (e.g. “heart cardiac problem”) and many of them overlap (e.g. the same disorder according to ICD, “NCI Clinical” and OPCS) while specific ICD codes for very relevant conditions are missing (e.g. aortic valve insufficiency, mitral valve prolapse, hypertrophic/dilated cardiomyopathy, atrial fibrillation ect. ect.)

In many cases the specific diagnoses would likely be more useful than the unspecific/general diagnoses. For example, atrial fibrillation associates with predicted PALMD expression in brain and esophagus in the S-PrediXcan analysis (ST14) but the phenotype “heart_arrhythmia” is flat in the UKB-pheWAS in both tissues. Furthermore, since CAVS variants at the PALMD locus have been associated with aortic root size, BAV and cardiac septal defects (PMID: 29511194) it would have been very relevant to include congenital heart defects and echocardiographic traits. I realize that in many cases these phenotypes are simply not available, but UK Biobank has a large atrial fibrillation sample set, why is that not included?

We acknowledge that some potentially relevant phenotypes were missing in the original pheWAS analysis, which was first developed as a broad tool including the most frequent phenotypes in the UK Biobank. To fill this gap, we added the following phenotypes: aortic

valve insufficiency, mitral valve prolapse, mitral valve insufficiency, atrial fibrillation, dilated cardiomyopathy, hypertrophic cardiomyopathy, congenital malformations of the circulatory system, congenital malformations of cardiac septa, ventricular septal defect, atrial septal defect and congenital malformations in various systems. Echocardiographic traits were not available in the UK Biobank. The number of phenotypes included is now 852. None of the new phenotypes reached statistical significance, even when considering other approaches to correct for multiple testing (see response to comment #2 below).

We modified the Methods section accordingly. The updated list of phenotypes is available in Supplementary Table 1 and results in Figure 3 (see Appendix), Supplementary Figures 2-3 (see Appendix) and Supplementary Tables 6, 7, 9-16.

2. Correcting for multiple testing: It could be argued that a conventional Bonferroni correction is too conservative in the case of the UKB-pheWAS, in particular since many of the phenotypes overlap (as pointed out in #1) and are likely highly correlated. Would a less stringent approach, such as false discovery rate procedure be more appropriate? This is especially important when the main conclusion is about lack of association.

We now provide results with p-values adjusted using less stringent approaches to correct for multiple testing: false-discovery rate (Benjamini & Hochberg) and Benjamini & Yekutieli. In the pheWAS analyses, these approaches did not change the interpretation of the results (i.e. no phenotype reached statistical significance except for aortic valve related phenotypes for the genetically-determined expression in the aortic valve). In the S-PrediXcan analysis, a negative association between all ischemic stroke and predicted expression in the aortic valve was significant when using the false-discovery rate threshold. This association was in the same direction as aortic valve stenosis, suggesting another potential beneficial effect of increasing *PALMD* expression in the aortic valve.

We modified the Methods section accordingly:

*To correct for multiple testing, p-values were adjusted according to false-discovery rate (Benjamini & Hochberg), Benjamini & Yekutieli and Bonferroni methods. The number of tests was set to 852 in the pheWAS for genetically-determined expression of PALMD in the aortic valve and rs6702619. In the pheWAS for genetically-determined expression of PALMD in other tissues, the number of tests was set to 6,816 (8*852). In the eQTL analyses, the number of tests was set to 42,052 for the aortic valve and 1,147,088 for the other tissues (total number of transcripts). In the S-PrediXcan analyses, the number of tests was set to 189 (9*21). An adjusted p-value <0.05 was considered significant.*

All results in Supplementary Tables 6, 7, 9-17 now include p-values corrected for false-discovery rate (P_FDR) and Benjamini & Yekutieli (P_BY).

We added the following sentences in the Results and Discussion sections:

Using the false-discovery rate threshold, there was a significant association between PALMD genetically-determined expression in the aortic valve and ischemic stroke ($z=-3.1$, $P=0.0019$), with a low probability of colocalization ($PP4=15.7\%$) (Figure 5).

In addition to CAVS, a potential beneficial effect of increasing PALMD expression in the aortic valve was identified for ischemic stroke ($z=-3.1$, $P= 0.0019$).

3. Information on prediction models (ST5):

a. The calculations of predicted PALMD expression could be made more transparent to the reader. It would be very informing to list the variants underlying the calculations and their assigned weights, f.ex. in supplementary table 5.

b. The authors mention the low proportion of variance in expression (2-20%) explained in other tissues than the aortic valve as one of the limitations of the study. For a reader that is not knowledgeable about the details of PrediXcan (such as myself) this raises the question of what is considered high and low? Is there any consensus on what is an acceptable proportion for a particular model to be used to test for trait associations? Scanning the literature does not provide easy answers, if they exist it might be useful to provide the reader with some insight and a reference.

a. We added the list of variants and their respective weights in the models to predict gene expression in the 9 tissues (including the aortic valve) in Supplementary Table 3.

b. To our knowledge, there is no consensus regarding the minimum/ideal proportion of variance that should be explained by a model. The PredictDB pipeline considers a model significant when the Pearson correlation coefficient between predicted and observed expression in the validation dataset is greater than 0.1 (corresponding to $R^2 > 1\%$) and the estimated p-value for this statistic is less than 0.05. We added this information with the reference⁹ in the Methods section:

In the PredictDB pipeline, a model is considered significant if the average Pearson correlation coefficient between predicted and observed expression during nested cross validation is greater than 0.1 (equivalent to $R^2 > 0.01$) and the estimated p-value for this statistic is less than 0.05⁹.

4. A pheWAS using the lead CAVS variant at PALMD? This would likely be simple to perform in the same way as done for the predicted expression of PALMD (and as done for lead variants at three other CAVS loci here: PMID: 32141789). As the authors mention in the introduction, lead variants at the locus affect PALMD expression in the aortic valve and not in 48 other tissues in GTEx (although it says 44 tissues in the cited paper: PMID: 29511167, is this a mistake or has the analysis been repeated with additional tissues?). It would thus be interesting to know if a pheWAS using the lead variant is consistent with the pheWAS using the predicted expression score in aortic tissue.

June 19, 2020

We added a pheWAS using the lead CAVS risk variant rs6702619 (Supplementary Figure 2, see Appendix). As expected, the results are very similar to the ones for genetically-determined expression in the aortic valve.

We thank the reviewer for pointing out the discrepancy in the number of tissues. The original statement in PMID: 29511167² was based on GTEx v6p, which included 44 tissues. We now provide a cis and trans-eQTL analysis in the 48 tissues available in GTEx v7 (see also response to Reviewer 1 Minor points #2 and #3).

We corrected the statement in the Introduction and added the following in the Results section:

Of note, the variants associated with CAVS do did not affect PALMD expression in 44 other tissues in the Genotype-Tissue Expression (GTEx) project.

Transcriptome-wide eQTL analyses for rs6702619 in 48 tissues included in the GTEx Project v7 European samples as well as in our aortic valve expression dataset confirmed the known signal for PALMD expression in the aortic valve. No other association was significant following correction for multiple testing (Supplementary Table 8).

5. **Wording of conclusions:** All in all, there are several limitations to the study, most of which the authors mention in the discussion section. Some improvements may be possible (as suggested in #1-4) but other shortcomings seem harder to avoid, most notably the lack of analysis in other cardiac and vascular tissues than the aortic valve. It is important to note, that despite the shortcomings the analysis is informing and based on quality data, the aortic root sample set and the UK Biobank extensive phenotypic dataset. However, it is my opinion that in light of the limitations, the wording in the discussion section is a bit strong, for example: “*The absence of a significant effect of predicted PALMD expression on a wide range of health conditions suggests that its modulation has likely a limited impact on other organs and systems.*” and “*There are no adverse effects predicted by a genetic increase of its expression in the aortic valve.*” I would suggest toning this down, or for example adding phrases like “according to data available to us”. Also the “*modest associations with atrial fibrillation and cardio-embolic stroke*” might be underestimated in light of the before mentioned limitations (i.e. no analysis in heart tissue, only 2-20% of variance in expression explained by models). Finally, I think it is worth mentioning as a limitation that the phenotype list is not complete, and that f.ex. phenotypes that have been associated with CAVS-variants at PALMD (PMID: 29511194, PMID: 28394258: ASD/VSD/echocardiographic measurements, see #1) were not available for analysis.

We agree with the reviewer’s points.

We adapted the following sentences in the Discussion section:

The absence of a significant effect of predicted PALMD expression on a wide range of health conditions according to available data suggests that its modulation has likely a limited impact on other organs and systems.

The effects could potentially be underestimated, considering the lower variance explained in these tissues (<10%).

There are no adverse effects predicted by a genetic increase of its expression in the aortic valve based on available data.

Echocardiographic measurements were not available in UK Biobank.

6. Sex-stratified analysis: Sex-stratified analysis was only performed for the association of genetically-determined expression of PALMD in the aortic valve with CAVS. It would be interesting to also perform sex-stratified analysis for the AF/stroke associations in other tissues if possible.

Unfortunately, we cannot perform sex-stratified analyses for the significant associations for atrial fibrillation and stroke found in the S-PrediXcan analyses since only summary statistics are available. The corresponding associations in the UK Biobank pheWAS are modest ($P > 0.001$), therefore sex-stratified analysis would not be informative.

7. Improved phenotypic description: A description of the terms “clinical - NCI” and “clinical” should be included in the methods section and/or table footnotes.

We added the definitions of the acronyms used in the Supplementary Tables legends.

NCI: non-cancer illness code (self-reported); ICD: International Classification of Diseases 10th Revision; OPCS: Office of Population Censuses and Surveys Classification of Surgical Operations and Procedures, 4th Revision; Clinical: phenotypes obtained from participants visits (questionnaire or examination)

8. Comment on the discussion about potential mechanisms responsible for the association between PALMD expression and CAVS: In the discussion section it says: *“The lead PALMD CAVS risk variant has also been associated with the presence of aortic stenosis in patients with a bicuspid valve, a congenital anomaly in which the aortic valve has two cusps instead of three. However, this condition is significantly more frequent in men²⁴ and therefore cannot explain the stronger association between genetically-determined PALMD expression and CAVS observed in women.”*

First, I do not agree with the above statement that just because BAV in general is more in common in men, PALMD expression levels could not be mediating a stronger risk of BAV (and therefore CAVS) in women. However, it might be worth discussing in this context that the aortic valve eQTL dataset is only based on individuals with tricuspid valves which is an argument against the notion that PALMD expression mediates risk of CAVS only by increasing risk of BAV. Furthermore, it might also be worth mentioning that one of the lead CAVS associated variants at PALMD has been associated with increased aortic root dimension (PMID: 28394258) and that this association remains after removing BAV cases from the analysis (PMID:

29511194).

We reworded this section of the discussion to take into account the reviewer’s comments. We agree that a sex-specific effect of *PALMD* expression on BAV risk cannot be completely excluded. We did show previously that *PALMD* lead variant had a similar association with bicuspid and tricuspid CAVS¹. The effect of the variant on aortic root dimension in individuals with TAV is also another argument against a BAV-specific effect. We also added a reference showing that aortic valves from women with CAVS have more fibrosis and less calcification for a similar stenosis severity compared with men, which could be related to the higher risk conferred by the *PALMD* locus in women.

We modified the following section in the Discussion:

*Aortic valves from women with CAVS were shown to have more fibrosis and less calcification for a similar stenosis severity compared with men¹⁰. Women could therefore theoretically be more susceptible to a potential increase in valve fibrosis. The lead *PALMD* CAVS risk variant has also been associated with the presence of aortic stenosis in patients with a bicuspid aortic valve (BAV), a congenital anomaly more frequent in men¹¹ in which the aortic valve has two cusps instead of three, and with aortic root diameter in patients with a tricuspid valve¹². We showed previously a similar effect on the risk of CAVS when restricting to patients with a tricuspid or a bicuspid valve¹, suggesting that the risk is not mediated through BAV.*

9. Comment on the following statement in the discussion: “*Interestingly, although *PALMD* is expressed in various tissues, we found that the association with CAVS was restricted to genetically-determined expression in the aortic valve. Accordingly, in the eight tissues from the GTEx project for which a model could be developed to predict *PALMD* expression, we found no significant association with CAVS.*” – it would be appropriate to mention that this complies with the authors previous study reporting that “The CAVS risk alleles and increasing disease severity are both associated with decreased mRNA expression levels of *PALMD* in valve tissues.” and that ...” rs6702619-*PALMD* association was not significant in 44 tissues from GTEx, suggesting that the change in expression is specific to aortic valve tissue”...and cite that paper at this point in the discussion (PMID: 29511167).

We thank the reviewer for this suggestion, which we implemented in the Discussion:

*This is concordant with our previous finding that CAVS risk alleles and increasing disease severity are both associated with decreased mRNA levels of *PALMD* in valve tissues².*

References

- 1 Thériault, S. *et al.* Genetic association analyses highlight IL6, ALPL, and NAV1 as three new susceptibility genes underlying calcific aortic valve stenosis. *Circulation: Genomic and Precision Medicine* **12**, 431-441 (2019).
- 2 Theriault, S. *et al.* A transcriptome-wide association study identifies PALMD as a susceptibility gene for calcific aortic valve stenosis. *Nature communications* **9**, 988, doi:10.1038/s41467-018-03260-6 (2018).
- 3 Liu, B., Gloudemans, M. J., Rao, A. S., Ingelsson, E. & Montgomery, S. B. Abundant associations with gene expression complicate GWAS follow-up. *Nature genetics* **51**, 768-769, doi:10.1038/s41588-019-0404-0 (2019).
- 4 Karczewski, K. J. *et al.* The mutational constraint spectrum quantified from variation in 141,456 humans. *Nature* **581**, 434-443, doi:10.1038/s41586-020-2308-7 (2020).
- 5 Delaneau, O. *et al.* A complete tool set for molecular QTL discovery and analysis. *Nature communications* **8**, 15452, doi:10.1038/ncomms15452 (2017).
- 6 Battle, A., Brown, C. D., Engelhardt, B. E. & Montgomery, S. B. Genetic effects on gene expression across human tissues. *Nature* **550**, 204-213, doi:10.1038/nature24277 (2017).
- 7 Mathieu, P. & Boulanger, M. C. Basic mechanisms of calcific aortic valve disease. *The Canadian journal of cardiology* **30**, 982-993, doi:10.1016/j.cjca.2014.03.029 (2014).
- 8 Karlsson Linnér, R. *et al.* Genome-wide association analyses of risk tolerance and risky behaviors in over 1 million individuals identify hundreds of loci and shared genetic influences. *Nature genetics* **51**, 245-257, doi:10.1038/s41588-018-0309-3 (2019).
- 9 Gamazon, E. R. *et al.* A gene-based association method for mapping traits using reference transcriptome data. *Nature genetics* **47**, 1091-1098, doi:10.1038/ng.3367 (2015).
- 10 Simard, L. *et al.* Sex-Related Discordance Between Aortic Valve Calcification and Hemodynamic Severity of Aortic Stenosis: Is Valvular Fibrosis the Explanation? *Circulation research* **120**, 681-691, doi:10.1161/circresaha.116.309306 (2017).
- 11 Siu, S. C. & Silversides, C. K. Bicuspid aortic valve disease. *Journal of the American College of Cardiology* **55**, 2789-2800, doi:10.1016/j.jacc.2009.12.068 (2010).
- 12 Helgadottir, A. *et al.* Genome-wide analysis yields new loci associating with aortic valve stenosis. *Nature communications* **9**, 987, doi:10.1038/s41467-018-03252-6 (2018).
- 13 Barbeira, A. N. *et al.* Exploring the phenotypic consequences of tissue specific gene expression variation inferred from GWAS summary statistics. *Nature communications* **9**, 1825, doi:10.1038/s41467-018-03621-1 (2018).

APPENDIX: Updated and new figures

Figure 2: Relationship between valve *PALMD* eQTL and GWAS association with CAVS

LocusCompare plot³ showing the relationship between valve *PALMD* eQTL and GWAS association with CAVS for variants located within 1 Mb of *PALMD*. Colocalization PP4=99.7%. GWAS association was obtained from a meta-analysis of QUEBEC-CAVS and UK Biobank. The lead GWAS variant is annotated.

Figure 3: PheWAS of *PALMD* genetically-determined expression in the aortic valve in the UK Biobank

Each triangle represents a different phenotype (n=852). Triangles pointing up and down are positive and negative associations with *PALMD* genetically-determined expression in the aortic valve, respectively. The pink horizontal line represents the threshold for significance after correcting for multiple testing ($P=0.05/852=5.9 \times 10^{-5}$). The blue horizontal line represents the threshold for nominal significance ($P=0.05$).

Figure 4: Association between *PALMD* genetically-determined expression in 9 tissues and 21 cardiovascular phenotypes

Analyses were performed using S-PrediXcan¹³ and summary statistics from available GWAS meta-analyses (**Supplementary Table 2**).

HDL: high-density lipoprotein; LDL: low-density lipoprotein; CKD: chronic kidney disease; eGFR: estimated glomerular filtration rate; BMI: body-mass index.

* $P_{\text{FDR}} < 0.05$

Figure 5: Relationship between *PALMD* eQTL in selected tissues and GWAS association with atrial fibrillation and stroke

June 19, 2020

LocusCompare plots³ showing the relationship between *PALMD* eQTL and GWAS association with atrial fibrillation and stroke for variants located within 1 Mb of *PALMD*. **A** Esophagus muscularis and atrial fibrillation, colocalization PP4=97.0%; **B** Subcutaneous adipose tissue and cardio-embolic stroke, colocalization PP4=5.4%; **C** Aortic valve and all ischemic stroke, colocalization PP4=15.7%; GWAS associations were obtained from Nielsen et al. and the MEGASTROKE Consortium. The lead GWAS variant is annotated.

Supplementary Figure 2: Phenome-wide association study of rs6702619-G in the UK Biobank

Each triangle represents a different phenotype (n=852). Triangles pointing up and down are positive and negative associations with rs6702619-G, respectively. The pink horizontal line represents the threshold for significance after correcting for multiple testing ($P=0.05/852=5.9 \times 10^{-5}$). The blue horizontal line represents the threshold for nominal significance ($P=0.05$).

Supplementary Figure 3: Phenome-wide association studies of PALMD genetically-determined expression in 8 tissues in the UK Biobank

A) Brain anterior cingulate cortex

B) Transformed fibroblasts

C) Gastroesophageal junction

D) Esophagus mucosa

E) Esophagus muscularis

F) Tibial nerve

G) Pancreas

H) Subcutaneous adipose tissue

June 19, 2020

Phenome-wide association study of *PALMD* genetically-determined expression in **A** Brain anterior cingulate cortex; **B** Transformed fibroblasts; **C** Gastroesophageal junction; **D** Esophagus mucosa; **E** Esophagus muscularis; **F** Tibial nerve; **G** Pancreas; **H** Subcutaneous adipose tissue. Each triangle represents a different phenotype (n=852). Triangles pointing up and down are positive and negative associations with *PALMD* genetically-determined expression in the respective tissue. The pink horizontal line represents $P=0.05/852=5.9\times 10^{-5}$. The blue horizontal line represents the threshold for nominal significance ($P=0.05$).

REVIEWERS' COMMENTS:

Reviewer #2 (Remarks to the Author):

I thank the authors for their thorough replies to my comments. I think that in general the authors have addressed the key issues appropriately and their changes have improved the paper.

I have just one follow-up suggestion/consideration regarding the PALMD expression-AF/stroke associations and their interpretation:

The following statements seem to contradict each other, especially since AF is a common cause of ischemic/cardioembolic stroke:

- a) therapy increasing PALMD expression globally could cause "potential increase in risk of atrial fibrillation and cardio-embolic stroke"
- b) a "potential beneficial effect of increasing PALMD expression in the aortic valve was identified for ischemic stroke ($z=-3.1$, $P= 0.0019$)".

I realize that presumably some complex mechanisms might explain opposing effects of PALMD expression levels on stroke risk in different tissues.

However, in my opinion, the locusCompare analysis, which was suggested by the other reviewer, is very informing in this regard. It does support the conclusions about potential AF risk associated with increased PALMD expression, but does not support the claims about stroke. As the authors state in the manuscript, the probability of colocalization was low for stroke and PALMD expression, both in adipose tissue (PP4=5.4%) and aortic valve tissue (15.7%) - thus the colocalization analysis does not strongly support a causal relationship between PALMD expression and risk of stroke. Furthermore, from looking at Figure 5 it is quite clear that the top stroke variants are not the top PALMD eQTL variants, neither in adipose tissue nor the aortic valve. In fact, the eQTL and stroke associations in Fig5b are very weak.

In other words, I suggest skipping the statement of potential beneficial effects of increasing PALMD expression for ischemic stroke, and also about the association with stroke risk in adipose tissue. Instead just explain that the colocalization analysis does not support a causal relationship between PALMD expression and stroke. I believe this approach is more in line with the results and avoids contradictory statements.

Reviewer #3 (Remarks to the Author):

In the revised version of the manuscript, the authors have performed additional experiments and have addressed most of the comments. Only remaining major comment is with regard to fine-mapping analysis. The authors stated that the previous analysis identified rs6702619, as the most likely causal variant at this locus. However, it seems that the previous fine mapping study was performed using functional annotations of GWAS loci. Considering that this study uses genetically predicted gene expression, a fine mapping approach of expression data (for example COLOC or FOCUS) would provide further insight into cis-regulation of the expression signal.

July 24, 2020

Response to reviewers COMMSBIO-20-0584 – Final revisions

Reviewer #2 (Remarks to the Author):

I thank the authors for their thorough replies to my comments. I think that in general the authors have addressed the key issues appropriately and their changes have improved the paper.

I have just one follow-up suggestion/consideration regarding the PALMD expression-AF/stroke associations and their interpretation:

The following statements seem to contradict each other, especially since AF is a common cause of ischemic/cardioembolic stroke:

a) therapy increasing PALMD expression globally could cause “potential increase in risk of atrial fibrillation and cardio-embolic stroke”

b) a “potential beneficial effect of increasing PALMD expression in the aortic valve was identified for ischemic stroke ($z=-3.1$, $P=0.0019$)”.

I realize that presumably some complex mechanisms might explain opposing effects of PALMD expression levels on stroke risk in different tissues.

However, in my opinion, the locusCompare analysis, which was suggested by the other reviewer, is very informing in this regard. It does support the conclusions about potential AF risk associated with increased PALMD expression, but does not support the claims about stroke. As the authors state in the manuscript, the probability of colocalization was low for stroke and PALMD expression, both in adipose tissue (PP4=5.4%) and aortic valve tissue (15.7%) - thus the colocalization analysis does not strongly support a causal relationship between PALMD expression and risk of stroke. Furthermore, from looking at Figure 5 it is quite clear that the top stroke variants are not the top PALMD eQTL variants, neither in adipose tissue nor the aortic valve. In fact, the eQTL and stroke associations in Fig5b are very weak.

In other words, I suggest skipping the statement of potential beneficial effects of increasing PALMD expression for ischemic stroke, and also about the association with stroke risk in adipose tissue. Instead just explain that the colocalization analysis does not support a causal relationship between PALMD expression and stroke. I believe this approach is more in line with the results and avoids contradictory statements.

We modified the manuscript according to the reviewers' comments. The apparent discrepancy in the results is likely due to tissue-specific mechanisms. For clarity, we modified the statements regarding stroke in the Discussion section:

The absence of a significant effect of predicted PALMD expression on a wide range of health conditions according to available data suggests that its modulation has likely a limited impact on other organs and systems. Among other phenotypes related to cardiovascular risk, there were modest associations with atrial fibrillation (strongest for brain tissue, $z=5.2$, $P=2.2 \times 10^{-7}$) and cardio-embolic stroke (subcutaneous adipose tissue, $z=3.7$, $P=0.00024$) when looking at

July 24, 2020

predicted expression in non-cardiac tissues. Notably, the signals showed a high probability of colocalization for atrial fibrillation but not for cardio-embolic stroke.

We modified the Abstract accordingly, which was also updated in response to the editorial requests.

Reviewer #3 (Remarks to the Author):

In the revised version of the manuscript, the authors have performed additional experiments and have addressed most of the comments. Only remaining major comment is with regard to fine-mapping analysis. The authors stated that the previous analysis identified rs6702619, as the most likely causal variant at this locus. However, it seems that the previous fine mapping study was performed using functional annotations of GWAS loci. Considering that this study uses genetically predicted gene expression, a fine mapping approach of expression data (for example COLOC or FOCUS) would provide further insight into cis-regulation of the expression signal .

The results of a colocalization analysis using coloc are presented in Figure 2. There is a strong colocalization between the GWAS signal and the expression of *PALMD* in the aortic valve (PP4=99.7%). The lead SNP in the CAVS GWAS is also the lead SNP in the eQTL dataset (rs6702619). *PALMD* is therefore the gene most likely to be involved. It was in fact identified using TWAS analyses published previously (PMID 29511167 and 32141789). The FOCUS approach is a variation of the original TWAS approach using Bayesian statistics to prioritize genes, it would most likely lead to the same conclusions. As for the fine mapping of the causal variant(s), expression data is not very informative, as there are several variants in linkage disequilibrium with a strong effect on both CAVS risk and *PALMD* gene expression in the aortic valve (as shown in Figure 2). We added a sentence in the limitations section to acknowledge this:

*The causal variant and the mechanism by which it affects *PALMD* expression and aortic valve stenosis remain to be validated in experimental models.*